# Acute Administration of Ethanol and of a D1-Receptor Antagonist Affects the Behavior and Neurochemistry of Adult Zebrafish

**DOI:** 10.3390/biomedicines10112878

**Published:** 2022-11-10

**Authors:** Tanya Scerbina, Robert Gerlai

**Affiliations:** 1Department of Psychology, University of Toronto, Mississauga, ON L5L 1C6, Canada; 2Council of Ministers of Education, Toronto, ON M4V 1N6, Canada; 3Department of Cell & Systems Biology, University of Toronto, Toronto, ON M5S 3G5, Canada

**Keywords:** alcohol abuse, alcoholism, ethanol, ethyl alcohol, dopamine, shoaling, zebrafish

## Abstract

Alcohol abuse represents major societal problems, an unmet medical need resulting from our incomplete understanding of the mechanisms underlying alcohol’s actions in the brain. To uncover these mechanisms, animal models have been proposed. Here, we explore the effects of acute alcohol administration in zebrafish, a promising animal model in alcohol research. One mechanism via which alcohol may influence behavior is the dopaminergic neurotransmitter system. As a proof-of-concept analysis, we study how D1 dopamine-receptor antagonism may alter the effects of acute alcohol on the behavior of adult zebrafish and on whole brain levels of neurochemicals. We conduct these analyses using a quasi-inbred strain, AB, and a genetically heterogeneous population SFWT. Our results uncover significant alcohol x D1-R antagonist interaction and main effects of these factors in shoaling, but only additive effects of these factors in measures of exploratory behavior. We also find interacting and main effects of alcohol and the D1-R antagonist on dopamine and DOPAC levels, but only alcohol effects on serotonin. We also uncover several strain dependent effects. These results demonstrate that acute alcohol may act through dopaminergic mechanisms for some but not all behavioral phenotypes, a novel discovery, and also suggest that strain differences may, in the future, help us identify molecular mechanisms underlying acute alcohol effects.

## 1. Introduction

Abuse of alcohol (ethanol, ethyl alcohol or EtOH) represents a large societal problem worldwide [1,2,3]. Treatment options for alcoholism are limited [4,5,6] by our incomplete understanding of the mechanisms underlying alcohol’s actions in the brain [7,8,9,10]. One factor that has been proposed to play roles in the development of chronic alcohol abuse is the initial response to acute alcohol exposure [11,12]. For example, people who better tolerate the acute effects of alcohol may be prone to drink larger amount of alcohol and may start to drink more often, and thus have a higher risk of developing chronic alcohol abuse and dependence [13,14]. Acute effects of alcohol depend upon a variety of factors, including genetic differences among people [14]. However, the neurobiological mechanisms underlying acute alcohol effects, or the genes involved in individual differences in responses to acute alcohol consumption, are not fully understood.

Numerous animal models have been proposed to facilitate discovery of such mechanisms [15,16,17]. The zebrafish is a relative novice in this research, nevertheless, it has been proposed to be a promising model organism [18,19,20]. There are several reasons for this. The zebrafish has been found to possess numerous evolutionarily conserved features, from the nucleotide sequence of its genes [21,22,23], through its neurotransmitter systems [24] to its behavior [25,26]. Thus, it is considered to be a translationally relevant model organism for the analysis of human brain function and dysfunction [26,27,28]. The translational relevance is further increased when zebrafish results are compared and combined with those obtained with rodents and humans, and thus common overlapping features and mechanisms are identified [29]. The zebrafish is also considered to represent a reasonable compromise between system complexity (it is a vertebrate) and practical simplicity (it is small, easy to breed and cheap to keep in large numbers) [29]. Last, the method of alcohol delivery (and delivery of several other drugs or compounds) can be achieved in a non-invasive manner by immersing the fish in the solution [30,31]. For these reasons, we decided to use zebrafish and investigate the effects of acute alcohol administration using this model organism.

Alcohol is a complex drug from the perspective of pharmacological properties, as it directly interacts with a large number of molecular targets, and indirectly affects an even larger number of biochemical processes and neurobiological mechanisms [7,8,9,10]. Ideally, systematic large-scale mutation screens, drug/small molecule screens or comprehensive transcriptome analyses may be performed to discover the details of the complex mechanisms and effects of acute alcohol administration in the brain of vertebrates. Although such screens or systematic comprehensive studies are technically feasible, up to this date they have not been performed with any model organism for the analysis of acute alcohol effects. Nevertheless, the zebrafish has been employed in such large-scale comprehensive screening approaches aimed at other phenotypes and biological questions, particularly in the field of embryology [32,33]. In this paper, we describe results that represent a proof of principle, providing the first unequivocal piece of evidence that acute alcohol effects can be mediated by the dopaminergic neurotransmitter system in zebrafish, implying that comprehensive screening may also be feasible for mechanistic analysis of the actions of alcohol using this translationally relevant model organism.

Given the complex pharmacological profile of alcohol, the questions of what phenotype should one study to uncover acute alcohol effects, and what mechanisms, biochemical pathways or molecular targets, should one investigate are not simple to answer. We have decided to study behavior as the primary endpoint of acute alcohol administration-induced effects. Behavioral analysis may allow one to detect functional changes in the brain without having to have *a priory* information about where and what exactly these changes may be [26,34]. Furthermore, previously, we and others have found acute alcohol administration to alter a variety of behavioral phenotypes. For example, similarly to what has been shown in humans [35], acute alcohol administration has been found to alter responding to social cues, i.e., to the sight of conspecifics in zebrafish too. For example, shoaling (group forming) is dose dependently impaired by acute alcohol administration in zebrafish [31,36,37]. Similarly, locomotor activity (distance swum) is also significantly affected by acute alcohol, with lower doses increasing and higher doses decreasing activity in zebrafish [37,38], findings corresponding well to those obtained with humans [39].

As for mechanisms, we decided to focus on the dopaminergic system. There are multiple reasons for this. Alcohol has been shown to engage reward pathways in mammals, including humans [40], and the dopaminergic system has long been known to play crucial roles in reward [41]. In zebrafish, acute alcohol administration has been shown to increase dopamine and DOPAC (the metabolite of dopamine) levels in the brain [24,31,42,43,44], and has also been shown to lead to brighter coloration, which normally can be seen during spawning or fighting among zebrafish [37], resembling the euphoria induced by acute alcohol in humans [45]. Furthermore, while alcohol has been shown to disrupt shoaling [31,46,47], sight of conspecifics has been found to be rewarding in learning tasks [48,49,50,51] and to increase dopamine and DOPAC levels in the brain of zebrafish [52,53]. Additionally, acute administration of a dopamine D1-receptor (D1-R) antagonist has been shown to disrupt shoaling [50]. What is not known, however, is whether the dopaminergic neurotransmitter system is actually involved in acute alcohol administration induced behavioral changes. The current study is aimed at answering this question.

Thus, our working hypothesis is that, although alcohol is known to engage a variety of mechanisms in the brain, its disruptive effects on shoaling (social behavioral responses) may be mediated by the dopaminergic system. To test this possibility, we conducted a study with a 4 × 3 × 2 between subject experimental design. That is, we acutely exposed each experimental zebrafish to one of four alcohol concentrations (ranging between 0% and 1% vol/vol) and one of three dopamine D1-R antagonist concentrations (ranging between 0 and 1 mg/L), and we conducted these experiments on two genetically distinct zebrafish populations (strains). The reason for using two strains was that prior studies indicated strain differences in shoaling [54] as well as in alcohol induced behavioral and neurochemical changes [36,55,56,57]. We measured a number of behavioral responses, including distance to animated images of conspecifics (an artificial shoal) as well as measures of general activity. In addition, we also analyzed levels of neurochemicals from whole brain extracts, including levels of dopamine, DOPAC, serotonin and 5HIAA (the metabolite of serotonin). We investigated whether alcohol and/or the D1-R antagonist had main effects on these phenotypes and whether these effects were interacting or additive. We reasoned that if acute alcohol administration-induced effects on shoaling responses are mediated via the dopaminergic system, we should see significant interaction between the effects of alcohol and the D1-R antagonist. If, however, the effects of these two drugs are independent of each other, we should find additivity, i.e., lack of significant interaction. We emphasize that combined application of acute alcohol administration and dopamine-receptor antagonism has not been conducted before, and thus the question of whether alcohol can exert its behavioral and neurochemical effects via the dopaminergic system has not been answered, the goal of the current study.

## 2. Materials and Methods

### 2.1. Animals and Housing

All zebrafish tested in this study were bred, raised and maintained in the Gerlai Zebrafish Facility of the University of Toronto Mississauga as described before [49]. Briefly, fertilized eggs were collected from multiple spawning zebrafish pairs and kept in small hatching tanks. At 5 days post-fertilization (dpf), the free-swimming fry were transferred to Aquaneering nursery tanks (Aquaneering Inc., San Diego, CA, USA) and fed artificial plankton (100 μm diameter fry food by Zeigler, Gardens, PA, USA). Starting at age 10 dpf, fry were fed freshly hatched Artemia salina nauplii, and from 15 dpf onward, a mixture of crushed tetramin tropical flakes (Tetra Co., Melle, Germany) and spirulina flakes (Jehmco Inc., Lambertville, NJ, USA). At this age the small fish were transferred to 3 L holding tanks placed on the Aquaneering system rack. The Aquaneering rack uses a recirculating filtration system with mechanical (sponge), biological (fluidized bed), chemical (activated carbon) filter components as well as a UV sterilizing unit. In addition, to further improve water quality, 10% of the water on this rack was replaced automatically once a day. The system water had a pH of 7, and salinity was maintained at 300 μS. This level of salinity was achieved by reverse osmosis filtration and reconstitution of salt concentration by adding Instant Ocean Sea Salt. Light cycle was maintained at 12 h dark and 12 h light using ceiling mounted fluorescent light fixtures with lights turned on at 7:00 h.

We employed two genetically distinct populations of zebrafish, from here onward referred to as “strains”: a genetically well-defined quasi-inbred strain, AB, and a genetically heterogeneous population we designate as SFWT (short-fin wild type). The reasons for choosing these two populations were as follows. AB is the most frequently employed strain in zebrafish research. It is a quasi-inbred strain with 80% of the loci in a homozygous form. Its widespread use in zebrafish research facilitates replicability and reproducibility across laboratories [58]. As we were interested in whether genotype-dependent differences exist in alcohol and/or D1-R antagonist induced responses, we employed another zebrafish strain, SFWT. SFWT is a genetically heterogeneous population in which high genetic variance is expected among individuals, and high level of heterozygosity is expected across the genetic loci for each individual. This is because the SFWT zebrafish employed in this study are the first filial generation of parents (purchased from Big Als Aquarium Warehouse, Mississauga, ON, Canada) that came from a commercial breeding facility in Singapore, a location close to the natural geographic distribution of zebrafish, and in which a large number of zebrafish breeders are employed. We chose this population as we expect it to represent species typical features of zebrafish better than potentially unique inbred strains. All experimental zebrafish (AB and SFWT) were bred, raised and maintained in the same room of the Gerlai Zebrafish Facility under identical conditions and at the same time. All experimental zebrafish were tested when fully grown, sexually mature, at their age of 4–6 months post-fertilization.

### 2.2. Experimental Design, Drug Treatment Procedure

As briefly mentioned above, we employed a 4 × 3 × 2 between subject experimental design, with alcohol concentration having 4 levels (0.00 (control), 0.25, 0.50, 1.00 vol/vol% external bath), D1-receptor antagonist (SCH23390) having 3 levels (0.0 (control), 0.1, 1.0 mg/L external bath, and strain having 2 levels (AB, SFWT). For the behavioral analysis, sample size (n) of each of the 24 groups was 20, except for the following three groups: n(Strain AB, Alcohol 0.5%, D1-R antagonist 0.1 mg/L) = 21; n(Strain AB, Alcohol 1%, D1-R antagonist 0 mg/L) = 19; n(Strain AB, Alcohol 1%, D1-R antagonist 1 mg/L) = 21. For the analysis of neurochemicals sample size (n) of each of the group was 10. We chose these sample sizes because previously we have found significant behavioral and/or neurochemical changes induced by a variety of psychopharmacological studies we conducted, including those with alcohol and a D1-R antagonist [43,50].

Alcohol (anhydrous ethanol 99.95%) was obtained from the University of Toronto Med store. The dopamine D1-R specific antagonist SCH23390 (R-(+)-8-chloro-2,3,4,5-tetrahydro-3-methyl-5-phenyl-1H-3-benzazepine-7-ol) was obtained from Sigma-Aldrich, Oakville, ON, Canada (Cat # D054). Fish were randomly assigned to their respective treatment group, with each fish administered a given alcohol and D1-R antagonist only once (a between subject experimental design). The sex ratio for all groups was 50/50%, and sex was not found to have a significant main effect or interaction with any factors. Thus, data were pooled for sexes for all subsequent data analyses. Experimental zebrafish were first immersed in their respective D1-R antagonist solution for 30 min in a 300 mL beaker in which oxygenation was provided via an air-stone connected to an aquarium air-pump. Although ADME (absorption, distribution, metabolism and excretion) data are not available for most drugs in zebrafish, including for SCH23390, extrapolating from mammalian studies [59,60,61,62] as well as based upon previous zebrafish studies [43,50], we expected that 30 min long bath immersion in the above concentration of this D1-R antagonist should be sufficient to induce significant behavioral and neurochemical changes in adult zebrafish. Immediately following the immersion in the D1-R antagonist solution, zebrafish were placed in their respective alcohol bath concentration (also in 300 mL beakers equipped with oxygenating air-stone) for 60 min. The concentrations employed in this study correspond to prior concentrations of alcohol used acutely with adult zebrafish [31,36,38,43,44]. The length of immersion employed in the current study was based upon the previous finding showing that alcohol levels reached a steady maximal plateau in brain tissue after 60 min of bath immersion in adult zebrafish [63]. Furthermore, we expected alcohol to remain in the brain in sufficient amounts at least for the entire period of the 16 min long behavioral test session that immediately followed the immersion [63]. Immediately after the D1-R antagonist and alcohol bath immersion, zebrafish underwent behavioral testing. The temporal order of to which D1-R antagonist concentration and to which subsequent alcohol concentration zebrafish were exposed was randomized.

### 2.3. Behavioral Test and Procedure

The behavioral experimenter was blind to the strain origin (AB and SFWT appear identical) and to the bath concentration of D1-R antagonist and of alcohol the experimental zebrafish received. Furthermore, the temporal order of behavioral testing of the zebrafish followed the random temporal order of the drug concentrations. We employed a simple novel aquarium task described in a detailed manner previously [64]. Briefly, the experimental fish were placed singly in a 37.5 L glass aquarium (50 cm × 25 cm × 30 cm, length × width × height), flanked by an LCD computer monitor (17-inch Samsung SyncMaster 732 N) on each of its small sides. The aquarium was illuminated from above by a 15 W fluorescent light tube. The experimental fish were monitored and their behavior analyzed for a total of 16 min via a camcorder (JVC GZ-MG37u) that viewed the aquarium from the front. Video-recordings were transferred to a computer and later analyzed using Ethovision Color Pro Video-tracking software application (Version 3, Noldus Info Tech, Wageningen, The Netherlands). For the first 8 min of the recording session, no stimuli were presented to the experimental zebrafish. For the second 8 min of the session, computer animated images (moving with a varying speed from 1 to 4 cm/s) of 5 zebrafish of the same size as the experimental fish were presented on one of the LCD monitors flanking the experimental aquarium using a custom software application (ZFT, developed in-house and first described by Saverino & Gerlai [65] (also see [51]). The side on which these animated conspecific images were shown randomly varied between experimental zebrafish. Using the Ethovision software application, we extracted numerous parameters of the swim path of the experimental zebrafish as follows. To evaluate shoaling, the distance of the experimental zebrafish from the stimulus screen that presented the conspecific images was measured in cm [64]. It is the distance between the center-point of the experimental fish and the glass side wall of the experimental aquarium flanked by the active computer monitor. We also measured intra-individual (temporal) variance of the distance of experimental zebrafish from the glass wall flanked by the computer monitor that showed the images. This variance reflects how consistently/inconsistently the experimental fish behaved, i.e., how much they changed their distance on the horizontal axis relative to the location of the animated conspecific images. In order to evaluate locomotory activity, we measured the total distance the fish swum. Bottom dwell time is often regarded as a measure of anxiety [66]. To evaluate this response, we quantified the distance between the experimental fish and the bottom of the tank. We also quantified the intra-individual variance of distance from bottom, a measure of vertical exploration [67], i.e., variability of the location of the fish along the vertical axis. Last, we quantified absolute turn angle, i.e., the change of direction of locomotion from one frame to the next irrespective of whether it was clock- or counter-clockwise.

### 2.4. Evaluation of Levels of Neurochemicals Using HPLC

Previously, we found acute administration of alcohol to significantly increase the level of dopamine and its metabolite, DOPAC, as well as of serotonin and its metabolite 5HIAA [24,31,44,50]. We also found shoaling images to induce a rapid increase of dopamine and DOPAC levels, but no changes in serotonin or 5HIAA levels [52]. Thus, we decided to measure whether alcohol and/or the D1-R antagonist alters the levels of these neurochemicals independently (additive effects) or in combination (interaction), a question that has not been investigated before. Another reason for their choice was practical. They could be measured from the same tissue homogenates using a single run using high-precision liquid chromatography (HPLC) with electrochemical detection [24]. Our HPLC methods have been described in detail elsewhere [24]. Briefly, we exposed a set of AB and SFWT zebrafish to alcohol and the D1-R antagonist exactly the same way as those fish we tested behaviorally. Additionally, the fish we used for HPLC analysis were of the same size and age as those we studied in the behavioral experiment. Experimental zebrafish were quickly decapitated using surgical scissors and their brains were removed and placed on dry ice. Subsequently, brains were sonicated in artificial cerebrospinal fluid, and protein content of the sonicate was analyzed using a BioRad protein assay reagent (BioRad, Hercules, CA, USA). Next, the sonicate was centrifuged and the supernatant was collected for HPLC analysis. We employed a BAS 460 Microbore HPLC with electrochemical detection (Bio-analytical Systems Inc., West Lafayette, IN, USA) using a Uniget C-18 reverse phase microbore column as the stationary phase (BASi, Cat no. 8949). The mobile phase consisted of buffer (0.1 M monochloro acetic acid, 0.5 mM Na-EDTA, 0.15 g/L Na-octylsulfonate and 10 nM sodium chloride, pH 3.1, obtained from Sigma), acetonitrile and tetrahydrofuran (obtained from Fisher Scientific) at a ratio 94:3.5:0.7. The flow rate was set to 1.0 mL/min and the working electrode (Uniget 3 mm glass carbon, BAS P/N MF-1003) was set at 550 mV vs. Ag/Ag/Cl reference electrode. Detection gain was set to 1.0 nA, filter was at 0.2 Hz, and detection limit was set to 20 nA. Five μL of the sample supernatant was directly injected into the HPLC system for analysis. Standard dopamine, DOPAC, serotonin and 5HIAA (Sigma) were employed to quantify and identify the peaks on the chromatographs.

### 2.5. Data Calculation and Statistical Analysis

One may define social behavior as all behavioral responses that are induced by social stimuli. However, this may be a mistake. Social stimuli, such as the sight of animated conspecific images employed here, are expected to induce a variety of responses. For example, in addition to shoaling (swimming close to conspecifics) the sight of conspecifics may reduce anxiety/fear, i.e., antipredatory responses and thus, e.g., alter bottom dwelling [28,68,69,70], and/or may reduce exploratory behavior, e.g., locomotory activity measured by total distance swum and by amount of turning performed [64]. These three responses (shoaling, antipredatory responses and exploratory behavior) may represent independent behavioral strategies/responses/states elicited by social stimuli, and they thus may also have idiosyncratic underlying mechanisms. As a result, they may be influenced differently by alcohol and/or by the D1-R antagonist. To investigate this possibility, we decided to calculate the change in behaviors of zebrafish induced by the presentation of conspecifics, i.e., the temporal change from the pre-conspecific stimulus period to the conspecific-stimulus presentation period as follows. All behavioral measures were extracted from x-y coordinates measured 10 times a second using Ethovision Tracking software. The x-y coordinate data were first used to calculate the total distance swum, and the average of distance to stimulus, of variance of distance to stimulus, of distance to bottom, of variance of distance to bottom and of turn angle, respectively, recorded for the first 8 min and separately also for the second 8 min of the recording session. Then, we subtracted the thus calculated value obtained for each behavior for the first 8 min from the value obtained for that behavior of the second 8 min. That is, we calculated the change of behavior induced by the presence of animated conspecific images.

With our HPLC data, we conducted the following transformations before statistical analysis. We standardized the neurochemical amounts measured to the brain tissue sample weight. That is, we expressed our data as μg of neurochemical per mg of total brain protein.

Using SPSS (version 23 for the PC) we conducted three-factorial Variance Analyses (ANOVAs) to investigate whether the concentration of alcohol (factor 1), the concentration of D1-R antagonist (factor 2) and whether the strain origin of the fish (factor 3) had significant main effects on the above-described behavioral changes and on the relative neurochemical amounts, and whether there were significant interactions among/between these factors. Subsequently, we performed a Tukey Honestly Significant Difference (HSD) *post hoc* test for each strain separately, which allowed us to compare all 12 groups per strain with each other without inflating type-1 error. Although this type of *post hoc* test is excellent for determining whether particular groups or a group differ from others or another, given the large number of these groups in our study and given that not all group comparisons are informative, we focused on the pattern of results, and only present/discuss the most important significant differences among these groups identified by the Tukey HSD tests. 

We rejected the null hypothesis (no significant effect/interaction or no difference between/among groups) when the probability of such null hypotheses was not larger than 5%, i.e., when *p* ≤ 0.05. We note that the parametric statistical analyses we conducted here are insensitive to the violation of the criteria of homogeneity of variances and normality of distribution when the compared groups do not have grossly different sample sizes. Our sample sizes across the 4 × 3 × 2 (i.e., 24) treatment/strain groups were nearly identical. Thus, we did not check for the above criteria and did not perform scale transformations.

## 3. Results

### 3.1. Behavioral Parameters

Reduction of distance to the stimulus screen in response to the presentation of animated images of conspecifics is considered a measure of the strength of shoaling behavior [51]. Here, we also found the distance to stimulus screen to significantly diminish as a result of the presentation of animated images of conspecifics, as shown by the negative values presented on Figure 1. Although all treatment groups for both strains show this reduction of distance, the figure also demonstrates that the amount of reduction of the distance differed across the treatment groups and strains. For example, higher alcohol concentrations appeared to diminish this reduction, i.e., impaired the shoaling response. Similarly, D1-R antagonist also appeared to exert a dose dependent effect, reducing shoaling. Last, the effects of alcohol and the D1-R antagonist do not appear to be additive. These observations were confirmed by the results of ANOVA. It found the effect of alcohol (F(3, 457) = 13.91, *p* < 0.001), and the effect of the D1-R antagonist (F(2, 457) = 5.03, *p* = 0.007) to be significant, but the effect of strain non-significant (F(1, 457) = 2.79, *p* = 0.10). The alcohol x D1-R antagonist interaction was also significant (F(6, 457) = 4.32, *p* < 0.001), but the D1-R antagonist x strain (F(2, 457) = 1.98, *p* = 0.14), the alcohol x strain (F(3, 457) = 1.46, *p* = 0.23) and the alcohol x D1-R antagonist x strain (F(6, 457) = 0.96, *p* = 0.45) interactions were all non-significant. Post hoc Tukey HSD test confirmed a significant dose dependent alcohol effect. For example, among the 0 mg/L D1-R antagonist treated fish, the highest alcohol concentration (1%) group fish showed significantly (0.05%) less shoaling response compared to the control (0%) alcohol treated fish and to those treated with the intermediate concentrations (0.5 or 0.25%) in both strains. The significant interaction between alcohol and D1-R antagonist treatments is also well illustrated if one compares the D1-R dose response of 0% alcohol treated fish (first three bars on the bar graphs of Figure 1) with the D1-R antagonist dose response of the 1% alcohol treated fish (last three bars of Figure 1). Tukey HSD revealed that for the 0% alcohol treated AB zebrafish increasing D1-R concentrations had significant (*p* < 0.05) shoaling response reducing effect. Whereas for the 1% alcohol treated AB zebrafish, this dose response was reversed, albeit without significant (*p* > 0.05) differences among the D1-R concentration groups. The findings were similar, but not identical, for SFWT zebrafish, where the D1-R antagonist dose response was found U-shaped in the 0% alcohol treated SFWT fish (with the 0.1 mg/L and 1 mg/L D1-R concentrations group differing significantly, *p* < 0.05) but quasi-linear for the 1% alcohol treated SFWT fish (with the 0 mg/L and 1 mg/L D1-R antagonist treated fish differing significantly, *p* < 0.05).

Intra-individual variance in the shoaling response (the variance of the change of distance from the no-stimulus period to the stimulus period) is shown in Figure 2. This variance quantifies how consistently/inconsistently the fish responded to the presentation of the animated conspecific images. Negative values shown on this figure mean that the variance was reduced by the shoaling images, i.e., the experimental fish remained at a more constant distance from the stimulus screen upon the presentation of the conspecific images compared to how they behaved before the images were shown. Figure 2 also suggests that alcohol reduced this variance (consistency) in a dose dependent manner. Similarly, the D1-R antagonist also reduced the variance with high doses having the strongest effects. Last, it appears that AB responded with more reduction of variance (increased their consistency of how far they swum from the images) in response to the animated conspecific images than SFWT did. These observations were confirmed by ANOVA. It showed that the effect of alcohol (F(3, 457) = 18.23, *p* < 0.001), the effect of the D1-R antagonist (F(2, 457) = 5.20, *p* = 0.006) and the effect of strain were all significant (F(1, 457) = 7.76, *p* = 0.006). However, no interaction term was found significant (alcohol x D1-R antagonist (F(6, 457) = 0.23, *p* = 0.71), alcohol x strain (F(3, 457) = 0.96, *p* = 0.41), D1-R antagonist x strain (F(2, 457) = 0.64, *p* = 0.53), alcohol x D1-R antagonist x strain (F(6, 457) = 0.90, *p* = 0.49). Tukey HSD test confirmed that among the 0 mg/L D1-R antagonist treated zebrafish of both strains, exposure to the intermediate concentration of 0.25% alcohol reduced the variance of distance most (significantly (*p* < 0.05) more compared with other concentration groups) and the 1% alcohol treated zebrafish reduced their variance the least (significantly (*p* < 0.05) less than fish given 0.25% or 0% alcohol). Tukey HSD test also showed that the highest concentration of D1-R antagonist (1 mg/L) had the strongest effect leading to the least amount of variance reduction (significantly (*p* < 0.05) less compared to fish treated with 0 mg/L D1-R antagonist). 

Exploratory activity may also change when shoal mates are present, as their presence may diminish the need to find escape routes, search for predators or hiding places, and because food may also be more efficiently found by more eyes and lateral lines, i.e., by shoal-mates. Thus, although exploratory behavior is not a form of social behavior, it may alter in response to the sight of conspecifics. A measure of exploratory behavior is general locomotory activity. Here, we quantified total distance swum and expressed the values as the change of this distance induced by the images of conspecifics (Figure 3). Overall, the figure suggest that experimental zebrafish of both strains reduced their total distance in response to the shown images, but this reduction depended on alcohol and D1-R antagonist dose employed, with higher concentrations of these drugs diminishing the effect of the social stimuli (reduced change of distance). It also appears that AB zebrafish reduced their distance in response to the presentation of animated conspecific images more compared to SFWT zebrafish. These observations were confirmed by ANOVA, which showed that the effect of alcohol (F(3, 457) = 28.12, *p* < 0.001), the effect of the D1-R antagonist (F(2, 457) = 6.45, *p* = 0.002) and the effect of strain were all significant (F(1, 457) = 5.36, *p* = 0.02). However, no interaction term was found significant (alcohol x D1-R antagonist (F(6, 457) = 0.51, *p* = 0.80), alcohol x strain (F(3, 457) = 1.07, *p* = 0.36), D1-R antagonist x strain (F(3, 457) = 0.18, *p* = 0.83), alcohol x D1-R antagonist x strain (F(6, 457) = 1.35, *p* = 0.23). Post hoc Tukey HSD test showed that the highest alcohol dose group (1%) showed the least reduction of total distance (for example, among the 0 mg/L D1-R antagonist treated fish, the 1% alcohol treated fish significantly (*p* < 0.05%) differed from fish of all other alcohol groups in both strains). It also showed the fish treated with the highest D1-R antagonist dose (1 mg/L) also reduced their distance the least.

The pattern of results was quite different for the behavioral measure distance to bottom. Here, again, we assumed that the appearance of conspecific images should alter the distance experimental fish swim from the bottom. Bottom dwell time has been suggested as a measure of anxiety [66], and presence of conspecifics has been shown to decrease anxiety [68]. Thus, we assumed that presentation of conspecifics should increase the distance to bottom. Figure 4 shows this is not what we found. The generally negative values shown on this figure suggest that zebrafish of most treatment groups decreased their distance, i.e., swam closer, to the bottom in response to the presentation of animated conspecific images. ANOVA found that alcohol had a significant effect on this response (F(3, 457) = 4.67, *p* = 0.003), but the effect of the D1-R antagonist was non-significant (F(2, 457) = 1.76, *p* = 0.17). The strain effect was found significant (F(1, 457) = 9.01, *p* = 0.003). However, no interaction term was found significant (alcohol x D1-R antagonist (F(6, 457) = 1.57, *p* = 0.15), alcohol x strain (F(3, 457) = 0.57, *p* = 0.63), D1-R antagonist x strain (F(3, 457) = 0.35, *p* = 0.70), alcohol x D1-R antagonist x strain (F(6, 457) = 1.05, *p* = 0.39). Tukey HSD test showed that the apparent differences seen among alcohol and D1-R treatment groups are non-significant (*p* > 0.05), except that the highest alcohol concentration (1%) treated fish changed their distance from bottom less compared to the lower-intermediate alcohol concentration (0.25%) treated fish.

Intra-individual temporal variance of distance to bottom reflects how much the fish move up and down in the water column, i.e., it is a measure of vertical exploration. Exploration is expected to diminish when fish become habituated to a novel environment, i.e., when the fish is experiencing less novelty induced anxiety [71]. As presence of conspecifics has been found to reduce anxiety [68], we expected the variance of distance to bottom to diminish in response to the presentation of animated conspecific images. This is exactly what we found, as demonstrated by the generally negative values (reduced variance) shown in Figure 5. From this figure it is also apparent that alcohol had a dose dependent effect, it diminished the reduction of variance to bottom. Although less obviously, but the D1-R antagonist also appeared to diminish this change. ANOVA found the alcohol (F(3, 457) = 21.64, *p* < 0.001) and the D1-R antagonist effects significant (F(2, 457) = 5.76, *p* = 0.003). The strain effect was non- significant (F(1, 457) = 0.42, *p* = 0. 52), and we found the interaction terms also non-significant (alcohol x D1-R antagonist (F(6, 457) = 1.27, *p* = 0.27), alcohol x strain F(6, 457) = 1.57, *p* = 0.20), D1-R antagonist x strain (F(3, 457) = 0.12, *p* = 0.89), alcohol x D1-R antagonist x strain (F(6, 457) = 0.92, *p* = 0.48). Tukey HSD test found that the highest alcohol concentration essentially abolished (*p* < 0.05) the change in vertical exploration induced by the animated conspecific images) irrespective of the D1-R antagonist dose employed.

The last behavior we measured and analyzed was absolute turn angle, i.e., the amount of turning irrespective of its direction. Although not described before, we expected turning to increase in response to the animated conspecific images, as we expected the experimental fish to follow these images near the short side of the aquarium swimming back and forth. This is exactly what we found. The increased turn angle is well demonstrated by Figure 6, which shows values for most treatment groups to be positive (increase from the pre-stimulus period to the stimulus period). Furthermore, the highest alcohol dose (1%) appears to have abolished this increase and the D1-R antagonist also appears to have blunted the increase. ANOVA confirmed these observations and found the effect of alcohol (F(3, 457) = 6.80, *p* < 0.001) and the effect of the D1-R antagonist (F(2, 457) = 4.47, *p* = 0.01) significant. The strain effect was non-significant (F(1, 457) = 2.05, *p* =0.15). The double interaction terms were also non-significant (alcohol x D1-R antagonist F(6, 457) = 0.27, *p* = 0.95; alcohol x strain F(6, 457) = 1.09, *p* = 0.35; D1-R antagonist x strain F(3, 457) = 0.27, *p* = 0.77). However, the triple interaction term, alcohol x D1-R antagonist x strain, was significant (F(6, 457) = 2.42, *p* = 0.026). Tukey HSD test showed that in AB zebrafish, the highest alcohol concentration significantly (*p* < 0.05) blunted the turn angle increasing effect of the animated conspecific images, and a similar significant effect of the highest dose of the D1-R antagonist was also found (*p* < 0.05). However, this significant alcohol effect was absent in SFWT fish.

### 3.2. Neurochemical Parameters

We next analyzed the levels of neurochemicals, dopamine, DOPAC, serotonin and 5HIAA from whole brain extracts using HPLC. Dopamine levels (Figure 7) appeared to be elevated by acute exposure to alcohol. In AB strain zebrafish, alcohol had a linear dose dependent effect with the highest dose having the strongest effect. The D1-R antagonist, on the other hand, reduced dopamine levels when applied at the highest concentration. It is also apparent that these two drugs did not have an additive effect (i.e., subtractive effect in this case) on dopamine levels. Another clearly observable finding is that both the absolute values and also the dose response curves for these drugs differ between AB and SFWT zebrafish. These observations were supported by the results of ANOVA. It found all main effects and interaction terms to be highly significant (alcohol F(3, 216) = 516.74, *p* < 0.001; D1-R antagonist F(2, 216) = 282.88, *p* < 0.001; strain F(1, 216) = 472.11, *p* < 0.001; alcohol x D1-R antagonist F(6, 216) = 14.53, *p* < 0.001; alcohol x strain F(6, 216) = 56.66, *p* < 0.001; D1-R antagonist x strain F(3, 216) = 7.64, *p* = 0.001; alcohol x D1-R antagonist x strain F(6, 216) = 10.59, *p* < 0.001). Tukey HSD tests confirmed these findings and, for example, found the effect of alcohol on AB zebrafish exposed to 0 mg/L D1-R antagonist linearly dose dependent with every dose group significantly (*p* < 0.05) different from the other, and also every alcohol dose group exposed the 0.1 mg/L D1-R antagonist from each other. However, for the 1 mg/L D1-R antagonist exposed AB zebrafish the alcohol dose response was found non-linear, i.e., plateauing for the two highest alcohol concentrations. That is, these two highest alcohol concentration groups did not significantly differ from each other (*p* > 0.05) but did significantly (*p* < 0.05) differ from the group treated acutely with the lowest alcohol concentration (0.25%) and also from the control group (0% alcohol). A somewhat different dose group differences were found by Tukey HSD for the SFWT zebrafish. That is, the SFWT zebrafish alcohol dose response curves were found inverted U-shaped and not linear when the fish were exposed to no D1-R antagonist (0 mg/L) or the lower dose of this drug (0.1 mg/L), with all alcohol concentration groups significantly (*p* < 0.05) differing from each other. Similarly to the AB fish, however, the SFWT zebrafish exposed to the highest D1-R antagonist dose (1 mg/L) the alcohol dose response showed a plateau, with the two highest alcohol concentration groups having the most similar values, both significantly (*p* < 0.05) higher than the values obtained for fish unexposed to alcohol (0% alcohol group) or the lowest alcohol dose (0.25% alcohol).

DOPAC levels (Figure 8) were also elevated by alcohol, with higher concentrations having stronger effects in both strains. Similarly to dopamine, the effect of the lower dose of the D1-R antagonist (0.1 mg/L) appeared to be negligible and did not appreciably modify the effect of alcohol. However, the higher dose of this drug (1 mg/L D1-R antagonist) had a clearly observable DOPAC reducing effect blunting the alcohol induced increase, with slightly different alcohol dose response trajectories in AB and SFWT fish. These results were supported by ANOVA, which found the effect of alcohol (F(3, 216) = 333.92, *p* < 0.001), the effect of D1-R antagonist (F(2, 216) = 106.61, *p* < 0.001), the effect of strain (F(1, 216) = 69.17, *p* < 0.001) all significant. The interaction terms alcohol x D1-R antagonist (F(6, 216) = 9.08, *p* < 0.001), alcohol x strain (F(6, 216) = 15.39, *p* < 0.001) were also found significant. However, the D1-R antagonist x strain interaction (F(3, 216) = 1.28, *p* = 0.28) and the alcohol x D1-R antagonist x strain interaction (F(6, 216) = 0.86, *p* = 0.53) were non-significant. Tukey HSD tests showed that in both AB and SFWT zebrafish increasing alcohol concentrations resulted in significantly elevated DOPAC with each alcohol concentration group differing from all others (*p* < 0.05) for fish that were exposed either to no D1-R antagonist (0 mg/L) or the lower concentration (0.1 mg/L). However, for fish that were exposed to the high concentration of D1-R antagonist (1 mg/L) Tukey HSD found all alcohol exposed fish (for AB) or the two highest alcohol concentration exposed fish (SFWT) not to differ significantly (*p* > 0.05) from each other but to significantly (*p* < 0.05) differ from the 0% alcohol group (AB), or the 0% and 0.25% alcohol groups (SFWT).

The alcohol and D1-R dose response curves obtained for the levels of serotonin (Figure 9), extracted from whole brain tissue samples appear to be different compared to those obtained for dopamine and DOPAC. It appears that alcohol only exerted a robust effect at the highest (1%) concentration in AB zebrafish, but in SFWT such robust effect was seen at the 0.5% alcohol concentration. Additionally, differently from the dopamine and DOPAC results, the D1-R antagonist did not appear to have a robust effect in any strain of zebrafish exposed to any alcohol dose. ANOVA confirmed a significant alcohol effect (F(3, 216) = 275.27, *p* < 0.001), but found the effects of D1-R antagonist (F(2, 216) = 0.22, *p* = 0.80) and strain (F(1, 216) = 0.82, *p* = 0.37) non-significant. The interaction term alcohol x D1-R antagonist reached the level of significance (F(6, 216) = 2.20 *p* = 0.04), and the alcohol x strain interaction was also significant (F(6, 216) = 305.21, *p* < 0.001). However, the D1-R antagonist x strain interaction (F(3, 216) = 0.05, *p* = 0.95) and the alcohol x D1-R antagonist x strain interaction (F(6, 216) = 0.79, *p* = 0.58) were non-significant. Tukey HSD showed that the D1-R antagonist was essentially ineffective as no significant differences (*p* > 0.05) among D1-R concentration groups were found for zebrafish within any given alcohol dose treatment set for either strain. As to the alcohol dose response curves, Tukey HSD confirmed that the 1% alcohol treated AB zebrafish significantly differed from all AB zebrafish alcohol groups, including the 0% alcohol control, all these other alcohol groups did not significantly differ from each other (*p* > 0.05). Tukey HSD test also confirmed that all 0.5% alcohol treated SFWT zebrafish significantly differed (*p* < 0.05) from SFWT zebrafish of all the other alcohol groups and these latter groups did not significantly differ from each other (*p* > 0.05).

The last result we consider is the effects of alcohol and of the D1-R antagonist on 5HIAA (Figure 10). For this metabolite of serotonin, alcohol appeared to exert a linear dose dependent effect in AB zebrafish, with higher alcohol concentrations elevating 5HIAA levels more strongly. Whereas for the SFWT fish the dose response curve was inverted U-shaped, with the strongest alcohol effect obtained for the 0.5% intermediate alcohol dose. The D1-R antagonist appeared to affect only certain alcohol dose groups. ANOVA found a significant alcohol (F(3, 216) = 166.07, *p* < 0.001), D1-R antagonist (F(2, 216) = 22.85, *p* < 0.001) and strain effect (F(1, 216) = 24.96, *p* < 0.001). The interaction term, alcohol x D1-R antagonist, was non-significant (F(6, 216) = 1.91 *p* = 0.08), but the alcohol x strain interaction (F(6, 216) = 40.30, *p* < 0.001), D1-R antagonist x strain interaction (F(3, 216) = 3.61, *p* = 0.03) and the alcohol x D1-R antagonist x strain interaction (F(6, 216) = 2.21, *p* = 0.04) were significant. Tukey HSD test confirmed the linear alcohol dose dependent effect in AB fish, i.e., found significant (*p* < 0.05) differences between groups of AB zebrafish receiving the increasing alcohol dose under the same D1-R treatment. For example, in AB, the effect of D1-R treatment was found significant (*p* < 0.05) for the 0% alcohol treated fish, as the 1 mg/L (highest dose) D1-R antagonist dose group differed from the 0 mg/L D1-R antagonist dose group. For the 1% alcohol treated AB fish, again the highest D1-R dose group was found to significantly differ from the no D1-R treated group. In SFWT, the D1-R antagonist was found to have an effect only under the 0% alcohol and 0.5% alcohol treatment. For the former, Tukey HSD found the 0 mg/L and the 1 mg/L D1-R antagonist treated fish to significantly differ (*p* < 0.05), and for the latter the 1 mg/L D1-R antagonist treated fish to significantly differ (*p* < 0.05) from both the 0 and the 0.1 mg/L D1-R antagonist treated groups.

## 4. Discussion

Zebrafish have previously been found to approach animated images of conspecifics upon their appearance and to stay in proximity of these images as long as they were being shown [64]. The strength of this shoaling response was found indistinguishable from that of responses elicited by live conspecifics or by video playback of previously recorded live conspecifics [72]. On the other hand, the response to animated conspecific images was clearly different from the response to animated images for which the pixels of the photos of conspecifics were scrambled and thus the images did not resemble a conspecific (or any other fish species) [52]. These scrambled images did induce a rapid approach (assumingly representing exploration of novel stimuli), but the experimental fish did not remain with these images, unlike when actual conspecific images were being shown [52]. Thus, we concluded that species-specific characteristics of the animated conspecific images must be present in order for experimental zebrafish to stay in close proximity to them [52,64,73]. In freely moving groups of zebrafish, members of the shoal were found to remain at a well-defined distance from each other, i.e., at a distance that is similar to how far a single experimental zebrafish swims from the animated images of conspecifics [73,74]. Thus, we concluded that the reduction of distance induced by animated images of conspecifics is an appropriate measure of the natural shoaling response of zebrafish [51]. In our current study, we found again that presentation of animated images of conspecifics reduced the distance between the experimental fish and the image presentation computer screen. In the past, we showed that acute exposure to alcohol disrupts shoaling in zebrafish [31,47]. This is what we observed in the current study too. At the highest dose (1%), acute alcohol almost completely abolished the shoaling response. We also knew from our past experiments that D1-R antagonism also disrupts shoaling [50], that conspecific images rapidly elevate dopamine and DOPAC levels in the brain of the exposed fish [52,53], and that acute alcohol also elevates dopamine and DOPAC levels [24,31,44]. We also discovered that acute alcohol increases tyrosine hydroxylase protein expression and dopamine synthesis in the zebrafish brain [42]. What we did not know is whether there is causal link between alcohol induced dopaminergic and behavioral responses. In the current study, we found proof for this causal link. 

We argued that if acute alcohol and the D1-R antagonist acts independently, we should be able to observe their effects on behavioral and neurochemical phenotypes as being additive, i.e., without significant interaction. Our results obtained for the shoaling response (change of distance to stimulus screen induced by conspecific images, Figure 1) demonstrates that the effects of acute alcohol and of the D1-R antagonist were non-additive, as we found a highly significant interaction between these two drugs. Importantly, the alcohol x D1-R antagonist interaction did not depend upon the strain origin of the fish, i.e., it was evident in both strains. For example, while in fish treated with 0% alcohol the D1-R antagonist dose dependently reduced the shoaling response, the trend was reversed in 1% alcohol treated fish, in which the highest D1-R antagonist dose treated fish showed the strongest shoaling response. The alcohol x D1-R antagonist interaction thus means that these two drugs do not exert their effects on the shoaling response via two independent (parallel) biochemical pathways or neurobiological mechanisms, but rather, alcohol and the D1-R antagonist converge on the same biological mechanism. Given that the D1-R antagonist employed here are thought to be a dopamine receptor specific drug in mammals and in zebrafish that thus influences the dopaminergic system [43,61], we conclude that acute alcohol affects shoaling via a dopaminergic mechanism in zebrafish. 

However, this finding does not mean that acute alcohol exerts its effects solely via dopaminergic mechanisms, or the effects of alcohol on all conspecific image induced behavioral responses are mediated by the dopaminergic system. For example, for the variance of the shoaling response (Figure 2, variance of the change of distance to the stimulus screen induced by the conspecific images), we found significant alcohol and D1-R antagonist effects, but these effects did not interact. First, we emphasize that “variance” here does not mean differences among individuals. Intra-individual variance here represents the time dependent changes in the distance of the single fish to the stimulus screen. It is, in essence, a measure of exploration of the conspecific images on the horizontal axis. High variance means the experimental fish often went closer and other times went further away from the conspecific images. The lack of alcohol x D1-R antagonist interaction, coupled with the significant main effects of these factors means that the effects of these two drugs acted independently on this measure of exploratory behavior. Alcohol increased the intra-individual temporal variance of shoaling response in a U-shaped dose dependent manner, with the highest variance induced by the low dose (0.25%) alcohol and the smallest variance by the highest dose (1%) alcohol. Whereas the D1-R antagonist reduced the intra-individual variance only if administered at the highest dose (1 mg/L). Again, importantly, these independent drug effects and dose response curves replicated well in both strains, AB and the SFWT. Thus, not only the lack of alcohol x D1-R interaction but also the trajectory of the dose response curves differed between the shoaling response and its intra-individual temporal variance (horizontal exploration). We speculate that such psychopharmacological differences between these two behavioral measures (the shoaling response and horizontal exploration behavior) may represent different underlying neurobiological mechanisms. Furthermore, because the effect of alcohol was independent of the effect of the D1-R antagonist, and because alcohol is known to act through a variety of mechanisms, we argue that the horizontal exploration behavior was influenced by alcohol likely via non-dopaminergic mechanisms.

To further strengthen this argument, one may examine the results we obtained for the behavioral measures, total distance swum or the intra-individual variance of distance from bottom. We first emphasize that again, just like in the above behaviors, we are discussing the change in these behaviors induced by the shoaling images. The statistical results as well as the dose response curves obtained for these two behavioral measures (Figure 3 and Figure 5) are practically identical to those discussed above for the horizontal exploration behavior (Figure 2). For example, the change in the total distance swum was affected significantly by alcohol and the D1-R antagonist, but these effects were independent of each other, i.e., were additive (subtractive in this case, to be precise). This is not surprising because distance traveled is often used as a measure of exploratory behavior in studies of fish [75,76,77] and mammals [78]. The intra-individual variance of distance from bottom, represents vertical exploration. Thus, we conclude that horizontal exploration (intra-individual variance of distance to the stimulus screen) vertical exploration (intra-individual distance to the bottom) and general exploratory behavior (total distance swum) are likely to be affected by alcohol and the D1-R antagonist via mechanisms that are, at least partially, different from those underlying shoaling behavior itself, and that alcohol affected these measures of exploratory behavior via non-dopaminergic mechanisms.

Examination of the results we obtained for the behavioral measure, distance to bottom reveals yet another different pattern of drug effects. Bottom dwell time is often regarded (although not without controversies) as a measure of anxiety in zebrafish [66]. Sight of conspecifics are expected to reduce anxiety and thus increase distance to bottom. We found the opposite, i.e., increased distance from bottom induced by the conspecific images, yet another indication that perhaps bottom dwell time, or distance from bottom, is not a reliable measure of anxiety in zebrafish [79,80]. Irrespective of whether distance to bottom is or is not an appropriate anxiety indicator, we found alcohol but not the D1-R antagonist to exert a significant effect on this response, yet again suggesting underlying mechanisms at least partially distinct both from those of shoaling and of exploratory behavior.

The last behavior we consider in this discussion is the change in turn angle. This behavioral response was affected significantly by both alcohol and the D1-R antagonist, but again not in interaction with each other. Additionally, importantly, the dose response curves of the drugs are unique for this behavior too: only the highest concentration of each of these two drugs had a significant effect in AB zebrafish, and only the highest concentration of the D1-R antagonist had an effect in SFWT, but alcohol was ineffective in this latter strain at all the concentrations examined.

We emphasize that all the above discussed behavioral effects reflect the change between the first and the second 8 min of the recording session, i.e., between the periods during which shoaling images were absent (first period) and the period during which they were present (second period). Although all these responses represent a change induced by the presentation of conspecifics, as we argued above, they may not all represent social behavior, i.e., shoaling. For example, while the reduction of distance to stimulus screen in response to conspecific images is indeed a shoaling response, alteration in vertical and horizontal exploration or distance traveled in general, represents changes in exploratory behavior. Our current findings of idiosyncratic drug effects showing, for example, alcohol x D1-R antagonist interaction for shoaling, but no such interaction and different dose response curves for exploratory behaviors, thus further strengthens the argument that these are distinct behavioral responses not just based upon their phenotypical appearance (and perhaps functional meaning), but also in terms of the mechanisms underlying them. Although these mechanisms are unknown at this point, our current results suggest that at least for shoaling, alcohol induced changes are mediated by the dopaminergic neurotransmitter system. Our neurochemical results are in line with this conclusion as they also revealed D1-R antagonist dependent alcohol effects.

Analysis of the effects of alcohol and the D1-R antagonist showed that both drugs significantly affected dopamine levels (Figure 7): alcohol significantly increased and the D1-R antagonist significantly decreased it. Importantly, the effects of these two drugs were not additive. A similar alcohol x D1-R antagonism interaction was also found for DOPAC (Figure 8), dopamine’s metabolite. Among the 0 mg/L and 0.1 mg/L D1-R antagonist treated AB zebrafish alcohol had a linear dose dependent effect but for the AB fish treated with the highest D1-R antagonist dose (1 mg/L) the alcohol effect plateaued at the 0.5% alcohol dose. A similar plateauing effect is also evident in AB fish for DOPAC. Such interactive effects of alcohol and the D1-R antagonist thus may explain the similar drug interaction found for the shoaling response. However, we also note the dose response trajectories obtained for dopamine and DOPAC (Figure 7 and Figure 8) do not precisely mirror those obtained for the shoaling response (Figure 1), suggesting that very likely the latter is also influenced by mechanisms other than the dopaminergic system.

In the past, we found shoaling images not to alter serotonin and 5HIAA levels in the brain of zebrafish [52]. We also found acute alcohol to increase the levels of these neurochemicals [31]. Importantly, here we show that although alcohol at the highest concentration in AB (1%), or at the intermediate concentration (0.5%) in SFWT, does increase serotonin levels, the D1-R antagonist had no effect. Given that here we found the D1-R antagonist to significantly affect shoaling but not serotonin levels, and given that shoaling images were previously not found to alter serotonin levels, we conclude that shoaling does not depend upon serotonin levels. Nevertheless, we cannot fully exclude that the serotoninergic neurotransmitter system influences alcohol induced changes in shoaling. This is because although the dose response trajectories for shoaling and for 5HIAA do differ, the levels of the latter neurochemical (the metabolite of serotonin) were influenced by alcohol and the D1-R antagonist in an interactive manner.

The last point we wish to discuss is the strain differences and the interactions of strain effect with alcohol and/or with the D1-R antagonist. In the current study, strain effects or strain x D1-R antagonist and/or strain x alcohol interactions were not found for the shoaling response but were found for other behaviors. While strain differences and strain interactions in these latter behaviors do complicate the interpretation of our results, as they mean that alcohol and/or D1-R effects are not faithfully replicated between the two strains, they also represent a potentially interesting starting point for future studies. Given that fish of the AB and SFWT strains were bred, raised, maintained and tested under identical conditions, the observed phenotypical differences between them must be the result of the genetic differences between the strains. Such genetic differences may be exploited in a variety of ways for the investigation of mechanisms underlying acute alcohol administration induced changes in brain function and behavior. For example, one may conduct a linkage analysis and identify the locus/loci of genes that may underlie differential alcohol effects, or, for example, one could also conduct gene expression analyses to identify differentially expressed genes involved in alcohol effects.

## 5. Conclusions

In this proof-of-principle study, we obtained the first piece of evidence that the shoaling response of zebrafish induced by the presentation of conspecific images is impaired by alcohol via the dopaminergic system. We also found that other behavioral responses to the shoaling stimulus are influenced by alcohol in a manner that is independent of dopaminergic mechanisms. Our study was not designed to pinpoint such mechanisms, or to essay the potentially large number of alcohol targets that could be involved. Nevertheless, given the powerful recombinant DNA methods and neuroscience techniques already developed for the zebrafish ([81]; for near comprehensive list of methods and applications also see [79]), our results highlight the possibility that this species may be an excellent model organism with which the mechanistic underpinnings of the effects of alcohol and of shoaling and other behaviors may be explored.

## Figures and Tables

**Figure 1 biomedicines-10-02878-f001:**
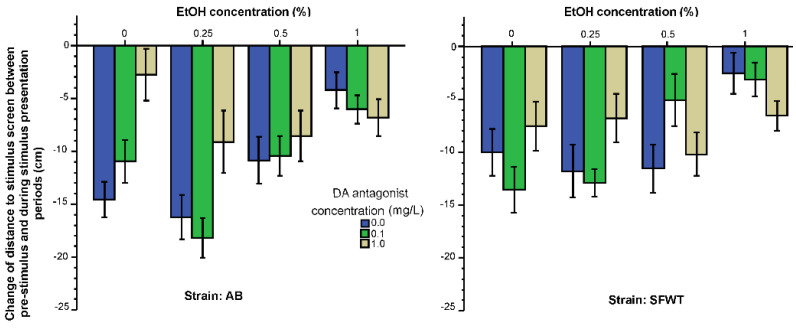
Change of distance to stimulus screen elicited by the presentation of animated conspecific images is influenced by the concentration of acute alcohol (X-axis) and the concentration of acute D1-R antagonist (legend) in an interactive manner. Mean ± S.E.M. are shown.

**Figure 2 biomedicines-10-02878-f002:**
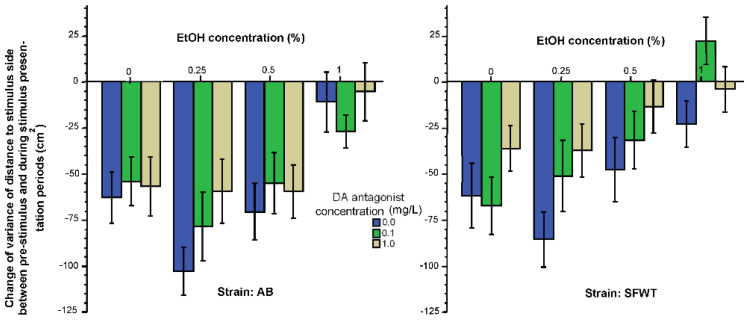
Change of temporal intra-individual variance of distance to stimulus screen (horizontal exploration) elicited by the presentation of animated conspecific images is influenced by the concentration of acute alcohol (X-axis) and the concentration of acute D1-R antagonist (legend) in an additive manner. Additionally, note the significant strain difference. Mean ± S.E.M. are shown.

**Figure 3 biomedicines-10-02878-f003:**
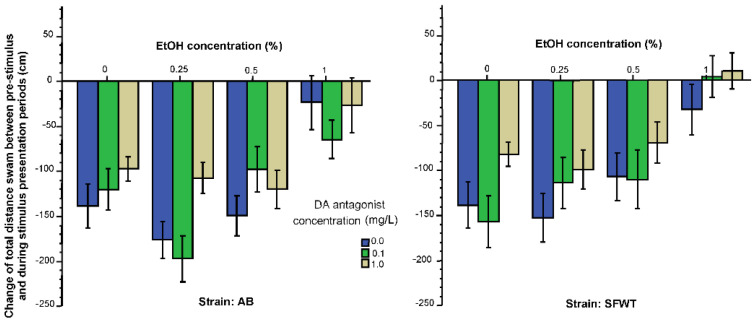
Change of total distance swum elicited by the presentation of animated conspecific images is influenced by the concentration of acute alcohol (X-axis) and the concentration of acute D1-R antagonist (legend) in an additive manner. Additionally, note the significant strain difference. Mean ± S.E.M. are shown.

**Figure 4 biomedicines-10-02878-f004:**
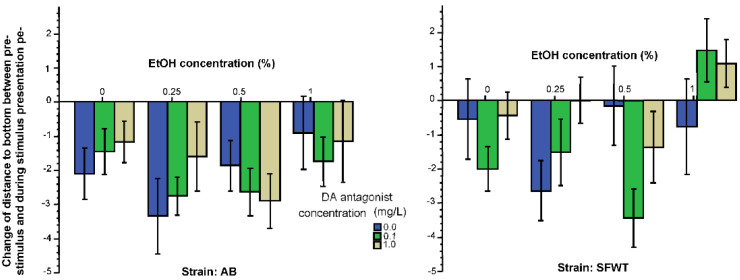
Change of distance to bottom elicited by the presentation of animated conspecific images is influenced by the concentration of acute alcohol (X-axis) and by the strain origin of the fish, but not by the concentration of acute D1-R antagonist (legend). Mean ± S.E.M. are shown.

**Figure 5 biomedicines-10-02878-f005:**
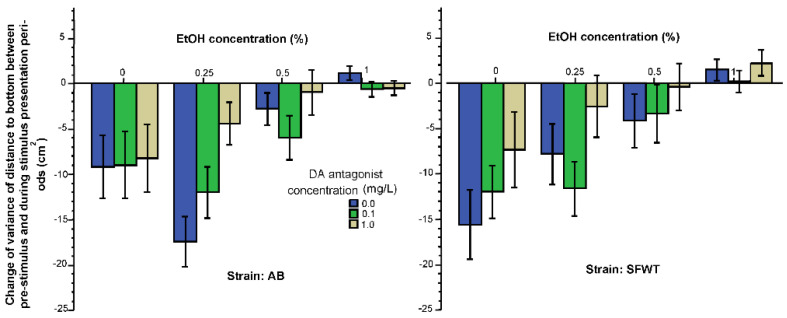
Change of intra-individual temporal variance of distance to bottom (vertical exploration) elicited by the presentation of animated conspecific images is influenced by the concentration of acute alcohol (X-axis) and by the concentration of acute D1-R antagonist (legend) in an additive manner. Mean ± S.E.M. are shown.

**Figure 6 biomedicines-10-02878-f006:**
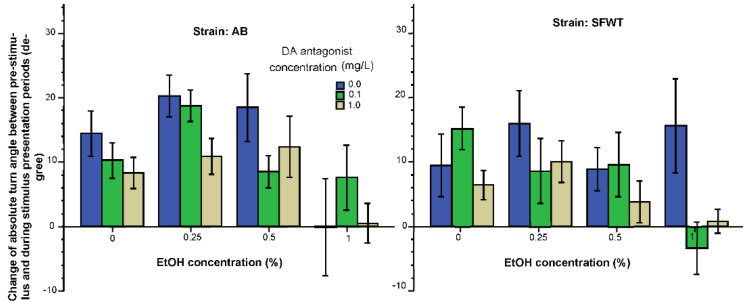
Change of absolute turn angle elicited by the presentation of animated conspecific images is influenced by the concentration of acute alcohol (X-axis) and by the concentration of acute D1-R antagonist (legend) in an additive manner. Mean ± S.E.M. are shown.

**Figure 7 biomedicines-10-02878-f007:**
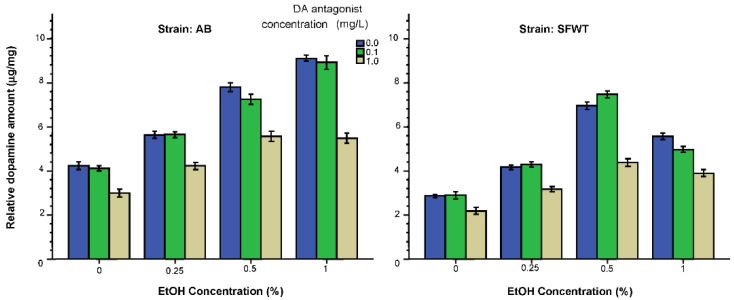
Dopamine levels relative to total brain protein weight are influenced by the concentration of acute alcohol (X-axis) and the concentration of acute D1-R antagonist (legend) in an interactive manner. Mean ± S.E.M. are shown. Additionally, note the significant strain difference.

**Figure 8 biomedicines-10-02878-f008:**
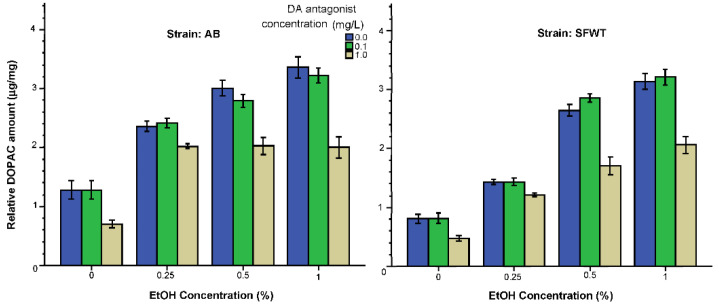
DOPAC levels relative to total brain protein weight are influenced by the concentration of acute alcohol (X-axis) and the concentration of acute D1-R antagonist (legend) in an interactive manner. Additionally, note the significant strain difference. Mean ± S.E.M. are shown.

**Figure 9 biomedicines-10-02878-f009:**
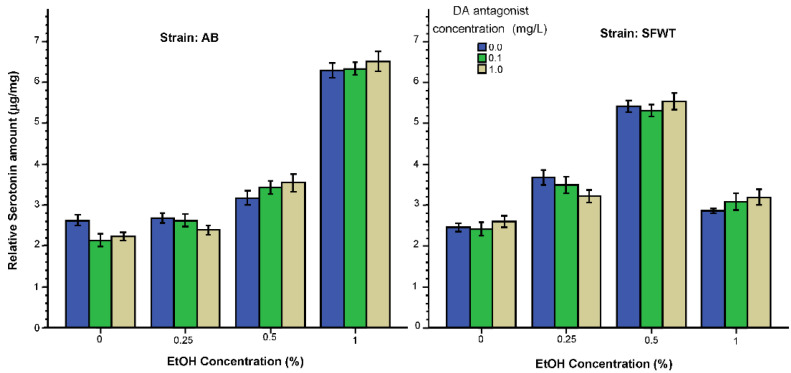
Serotonin levels relative to total brain protein weight are influenced by the concentration of acute alcohol (X-axis) but not by the concentration of acute D1-R antagonist (legend). Additionally, note the strain dependent alcohol dose responses (alcohol x strain interaction). Mean ± S.E.M. are shown.

**Figure 10 biomedicines-10-02878-f010:**
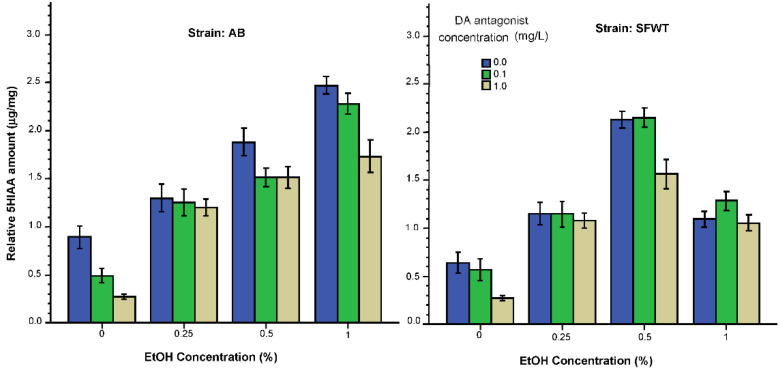
5HIAA levels relative to total brain protein weight are influenced by the concentration of acute alcohol (X-axis) and by the concentration of acute D1-R antagonist (legend). Additionally, note the strain dependent alcohol dose responses (alcohol x strain interaction). Mean ± S.E.M. are shown.

## Data Availability

The data present in this study are available from the corresponding author upon request.

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
