# Peer review of "Acute Administration of Ethanol and of a D1-Receptor Antagonist Affects the Behavior and Neurochemistry of Adult Zebrafish"

_biomedicines, 2022, doi:10.3390/biomedicines10112878_

Round 1

Reviewer 1 Report

This proof-of-principle manuscript, " Acute administration of ethanol and of a D1-receptor antagonist affects the behavior and neurochemistry of adult zebrafish" by Gerlai & Scerbina submitted to Biomedicines findings provided further evidence for using zebrafish as a model to examine alcohol neurobiological effects. The author's results indicated that zebrafish's social response (i.e., shoaling) appears to be mediated via the dopaminergic system. In contrast, exploratory behaviors do not appear to be dependent on the dopaminergic system. The authors also found that there were strain differences in exploratory behaviors. Overall, these findings are novel and interesting.

However, there are some concerns.

Introduction

1     1)  The authors do not need to mention the results in the introduction. Therefore, they should delete the last paragraph in the introduction.

Methods and Results

1   1)    Symbols should be added to graphs to indicate the significant findings between the groups.

2    2)    Page 12, line 530: This line says "the lowest alcohol dose (0.24% alcohol)" but the lowest dose is 0.25% for all groups. In figure 7, the x-axis has 0.2 instead of 0.25, like all the other graphs. The authors should double-check the manuscript for typos and grammar.

Author Response

******************

We thank the referee for reading our manuscript and for helping us improve it.  We have made most changes requested, and we list all these changes and responses to the comments in the order they were made by the referee.  Our changes and responses are in between lines of asterisks following each referee comment.

****************

This proof-of-principle manuscript, " Acute administration of ethanol and of a D1-receptor antagonist affects the behavior and neurochemistry of adult zebrafish" by Gerlai & Scerbina submitted to Biomedicines findings provided further evidence for using zebrafish as a model to examine alcohol neurobiological effects. The author's results indicated that zebrafish's social response (i.e., shoaling) appears to be mediated via the dopaminergic system. In contrast, exploratory behaviors do not appear to be dependent on the dopaminergic system. The authors also found that there were strain differences in exploratory behaviors. Overall, these findings are novel and interesting.

*****************

We appreciate this positive evaluation, thank you.

*******************

However, there are some concerns.

Introduction

1     1)  The authors do not need to mention the results in the introduction. Therefore, they should delete the last paragraph in the introduction.

*********************

Paragraph is deleted as suggested.  We replaced this last paragraph with a sentence that re-emphasized the main question the manuscript is intended to address as follows: “We emphasize that combined application of acute alcohol administration and dopamine-receptor antagonism has not been conducted before, and thus the question of whether alcohol can exert its behavioral and neurochemical effects via the dopaminergic system has not been answered, the goal of the current study.”

*********************

Methods and Results

1   1)    Symbols should be added to graphs to indicate the significant findings between the groups.

**************

We appreciate this request.  However, after careful consideration, we have rejected it.  The reasons are as follows.  With 12 groups to compare, showing pairwise group differences on the graphs would be impossibly complicated.  One could draw lines between each possible pairs of groups and indicate the significance level but this would lead to confusingly large number of lines and symbols.  One could use letter coding so that bars that share at least one letter would be considered non-significantly different (the preferred method for graphs of multiple range comparison type analyses), but even this would mean that we would have to use a large number of letters, too large to be intuitive and to guide understanding of which group is different from which.  Really, the only meaningful way significant vs non-significant differences could be shown is if we could reorder the bars according to their mean and then use lines underneath or above the bars showing non-significant ranges (basically graphically presenting what the Tukey HSD test does).  However, reordering would mean idiosyncratic rank order for each behavioral and neurochemical measure, which would be confusing, and would also make it impossible for the reader to observe the dose response curves for alcohol and the D1-R antagonist.  Thus, we decided to keep the order of the bars and structure our bar graphs according to alcohol concentration and D1-R antagonist concentration employed.  But this ordering does not allow indicating non-significant ranges.  We have consulted with statisticians and they suggested what we, at the end, did in the manuscript: describe the Tukey HSD results verbally in the text and focus on the most meaningful differences/lack of differences according to biology and/or working hypotheses.  We hope the referee will see our points and will agree that this is the clearest and most concise way of describing the results.

*****************

2    2)    Page 12, line 530: This line says "the lowest alcohol dose (0.24% alcohol)" but the lowest dose is 0.25% for all groups. In figure 7, the x-axis has 0.2 instead of 0.25, like all the other graphs. The authors should double-check the manuscript for typos and grammar.

********************

Thank you for spotting these typos.  They have now been corrected.

*********************

Reviewer 2 Report

The manuscript "Acute administration of ethanol and of a D1-receptor antagonist affects the behavior and neurochemistry of adult zebrafish" adds knowledge to the field and presents potentially interesting findings. Nevertheless, some questions should be addressed in order to improve its scientific quality:

- The article generally clear but contains a number of typos and grammatical errors - these should be corrected throughout.

- Titles of each subsection in the Results section should be substitute by a sentence stating the results obtained.

- It is difficult for me to understand what is the novelty of the results, since other similar articles have been published recently:

https://pubmed.ncbi.nlm.nih.gov/33667508/

https://pubmed.ncbi.nlm.nih.gov/30593828/

https://pubmed.ncbi.nlm.nih.gov/30172631/

Author Response

******************
We thank the referee for reading our manuscript and for helping us improve it.  We have made the changes requested and list them in the order they were made by the referee.  Our changes and responses are in between lines of asterisks following each referee comment.
****************

The manuscript "Acute administration of ethanol and of a D1-receptor antagonist affects the behavior and neurochemistry of adult zebrafish" adds knowledge to the field and presents potentially interesting findings. Nevertheless, some questions should be addressed in order to improve its scientific quality:

************

We appreciate this positive remark.

*************

- The article generally clear but contains a number of typos and grammatical errors - these should be corrected throughout.

******************

Thank you again, for the positive remark.  Regarding typos and grammar, we have combed through the manuscript and could only detect a few punctuation errors, and a couple of typos regarding alcohol dose values.  We have corrected them.  We did not detect other typos or grammatical errors.  One of us is a native English speaker and the other is a certified English as a Second Language (ESL) instructor, and thus we are confident our text is in a good shape.

*********************

- Titles of each subsection in the Results section should be substitute by a sentence stating the results obtained.

*****************

We have made this change as suggested.  There was one subtitle that referred to a specific behavior.  We have removed it.  For clarity, we now have two subsections/subheadings under “Results”: ‘3.1 Behavioral Parameters’, and ‘3.2 Neurochemical Parameters’.

******************

- It is difficult for me to understand what is the novelty of the results, since other similar articles have been published recently:

*********************

We have clarified this further in the revised manuscript.  For example, in the Abstract, we now write: “These results demonstrate that acute alcohol may act through dopaminergic mechanisms for some but not all behavioral phenotypes, a novel discovery.

The second part of the introduction we centered around clarifying what is new.  For example, we write: ”What is not known, however, is whether the dopaminergic neurotransmitter system is actually involved in acute alcohol administration induced behavioral changes. The current study is aimed at answering this question.  Thus, our working hypothesis is that, although alcohol is known to engage a variety of mechanisms in the brain, its disruptive effects on shoaling (social behavioral responses) may be mediated by the dopaminergic system.”  Also, we now end the intro with the following sentence: “We emphasize that combined application of acute alcohol administration and dopa-mine-receptor antagonism has not been conducted before, and thus the question of whether alcohol can exert its behavioral and neurochemical effects via the dopaminergic system has not been answered, the goal of the current study.”

Number of sections of the Discussion are also written to clarify what is new and why it is important to publish these novel findings.  For example, we state the following “What we did not know is whether there is causal link between alcohol induced dopaminergic and behavioral responses. In the current study, we found proof for this causal link.”  Also, in the Conclusion section we now clarify that “In this proof-of-principle study, we obtained the first piece of evidence that the shoaling response of zebrafish induced by the presentation of conspecific images is impaired by alcohol via the dopaminergic system.  We also found that other behavioral responses to the shoaling stimulus are influenced by alcohol in a manner that is independent of dopaminergic mechanisms.”

We thank the referee for suggesting the three studies by giving us their web-address from the pubmed system.  We have carefully read these publications and appreciate that the referee has drawn our attention to them. 

However, we note that these studies do not address the questions our study was designed to address.

For example, https://pubmed.ncbi.nlm.nih.gov/33667508/, the referee suggested we attend to, is a study on developmental toxicity induced by acute exposure to N-Ethylpentylone in zebrafish larvae.  It does not address how acute ethanol affects brain function and behavior.

Similarly, https://pubmed.ncbi.nlm.nih.gov/30593828/ is a study that does not address ethanol related questions at all, as this study is about the effects of triadimefon, a fungicide.

Last, https://pubmed.ncbi.nlm.nih.gov/30172631/, focuses on developmental alterations induced by D1-R and D2-R antagonism, but again the study does not address alcohol related questions.

In sum, while these above three suggested studies are all excellent and interesting, and were conducted with zebrafish, they do not address questions about alcohol related mechanisms, and certainly do not address the question whether acute alcohol can exerts its effects on behavior and neurochemical phenotypes via mechanisms associated with the dopaminergic system, a question we explored with our current study.

Round 2

Reviewer 1 Report

The authors have addressed the concerns.